# Isoflurane Rescue Schizophrenia-Related Deficits through Parvalbumin-Positive Neurons in the Dentate Gyrus

**DOI:** 10.3390/biomedicines10112759

**Published:** 2022-10-31

**Authors:** Hualing Peng, Jie Jia, Yisheng Lu, Hua Zheng

**Affiliations:** 1Department of Physiology, School of Basic Medicine, Tongji Medical College, Huazhong University of Science and Technology, Wuhan 430030, China; 2Institute of Brain Research, Collaborative Innovation Center for Brain Science, Huazhong University of Science and Technology, Wuhan 430030, China; 3Hubei Key Laboratory of Drug Target Research and Pharmacodynamic Evaluation, Huazhong University of Science and Technology, Wuhan 430030, China; 4Department of Anesthesiology, Tongji Hospital, Tongji Medical College, Huazhong University of Science and Technology, Wuhan 430030, China

**Keywords:** isoflurane, schizophrenia, adult neurogenesis, synaptic plasticity, parvalbumin-positive interneuron

## Abstract

The therapeutic effects of volatile anesthetics on mental diseases, particularly schizophrenia, have gained considerable interest. Although isoflurane is a commonly used volatile anesthetic, there’s no more evidence that it could work on treating schizophrenia. Here, we discovered that inhaling isoflurane at low concentrations might reverse the behavioral phenotypes of schizophrenia caused by MK801, such as hyperlocomotion, pre-pulse inhibition impairment, and working memory loss. Isoflurane also helped recovering adult neurogenesis and synaptic plasticity impairments in the dentate gyrus (DG) induced by MK801. To better understand the mechanism, we discovered that isoflurane could reverse the reduction of parvalbumin (PV)-positive GABAergic interneuron (PVI) number and the aberration of NRG1-ErbB4 signaling in the DG; however, isoflurane could not reverse the schizophrenia-related phenotypes caused by PVI ablation, indicating that PVI are necessary for the therapeutic effect of isoflurane. Interestingly, isoflurane could reverse phenotypes caused by blocking PVIs GABA release in the DG, indicating the therapeutic impact is independent of PVI GABA release. Our research revealed that isoflurane might be used to treat schizophrenia, possibly through PVI in the DG.

## 1. Introduction

Schizophrenia is a severe chronic mental illness that affects 1% of people worldwide from adolescence or early adulthood. As a result, it has a significant economic impact on both individuals and society [1,2]. Three types of schizophrenia symptoms can be distinguished: positive, negative, and cognitive. Antipsychotic medications can cure positive symptoms such as hallucinations and delusions, but they are almost never effective for treating cognitive problems [3]. Volatile anesthetics, such as nitrous oxide (N_2_O), isoflurane and sevoflurane, have recently drawn keen attention for their therapeutic effects on mental illnesses due to their easy administration, safety, and rapid onset of action [4,5,6]. Recent research suggests that inhaling sevoflurane at low concentration for several days can attenuate the symptoms of schizophrenia patients and schizophrenia-like phenotypes in mice, such as hyper-locomotion and social interaction deficit [7]. However, more research is necessary to determine whether volatile anesthetics can treat cognitive impairment, including working memory, and the mechanism of their therapeutic effects is still largely unknown.

Even though sevoflurane is widely used in clinical anesthesia, isoflurane, a halogenated ether substance and a positive allosteric modulator of the GABA_A_ receptor [8], might be more effective in treating mental diseases due to its higher blood-gas solubility than sevoflurane [9,10]. Adult neurogenesis impairment is crucial in mental illnesses, including schizophrenia and depression [11]. In the dentate gyrus (DG) of adult rats, anesthetic treatment, especially the inhalation of isoflurane can help to promote neural stem cell proliferation and maturation [12]. GABAergic interneurons, particularly the parvalbumin-positive interneurons (PVIs), tightly regulate the activity of developing neurons [13,14]. One of the primary pathophysiological mechanisms of schizophrenia is PVI dysfunction, and a brief dose of isoflurane inhalation can boost PVI activity [15]. However, it is yet unknown whether isoflurane can reverse schizophrenia-like phenotypes through PVI and consequently improve adult neurogenesis.

Here, we demonstrate that inhaling isoflurane at low concentrations for five days can reverse the MK801-induced schizophrenia-like behavioral abnormalities and adult neurogenesis deficit in the DG. The therapeutic effects of isoflurane appear to depend on PVIs in the DG. Intriguingly, GABA release from PVI is not required during the process. Our research sheds light on a novel therapeutic approach to treating schizophrenia and points to a neurological basis for the typical processes of volatile anesthetics.

## 2. Materials and Methods

### 2.1. Animals

All the in vivo surgery procedures were performed following the animal protocols approved by the Animal Welfare Committee of Huazhong University of Science and Technology. PV-Cre (stock No. 008069) mice were back-crossed for more than ten generations with C57BL/6J and maintained in groups (no more than five mice per cage) at a stable temperature (23–25 °C), exposed to a 12-h light/dark cycle with food and water provided ad libitum. All efforts were made to minimize suffering and the number of animals used.

### 2.2. Experimental Design and Drugs Administration

Mice (6-week-old) received intraperitoneal (i.p.) injections of 0.4 mg/kg MK801 (M107, Sigma-Aldrich, St. Louis, MO, USA) twice a day (at 10:00 a.m. and 3:00 p.m.) for five consecutive days to induce symptoms of schizophrenia. Then, mice were placed in a 25 × 20 × 15 cm plexiglass chamber, and 0.7% isoflurane was delivered via an isoflurane vaporizer at 1 L/min in 30% oxygen balanced with nitrogen for 1 h (10:00 a.m. to 11:00 a.m.) once per day during the five consecutive days (Figure 1A). The control group received identical gases without isoflurane. After behavioral tests, mice were sacrificed and the brain samples were subjected to electrophysiology recordings, Western blotting and immunofluorescence staining.

### 2.3. Stereotaxic Viral Injection

Adult mice (6 to 8-week-old) were anesthetized with pentobarbital sodium (50 mg/kg, i.p.) and head-fixed in a stereotaxic device (68025, RWD life science, Shenzhen, China). Viruses were bilaterally injected into the DG (0.4 µL per side, 30 nL/min) with a glass pipette (tip size, ~20 µm) at the following coordinates relative to bregma: anteroposterior, −2.05 mm; dorsoventral, 2.05 mm; and mediolateral, ±1.5 mm. After injection, the glass pipette remained in place for 15 min before the withdrawal. The titers of AAV-flex-tsCasp3-TEVp, AAV-CAG-DIO-EGFP-Tettox (GeneChem technologies) and ROV-U6-EF1α-EGFP-Flag (GeneChem technologies, Shanghai, China) were 10^12^ genome copies per mL. Brains were sectioned afterward to verify the infusion sites. Data were excluded if GFP expression was misdirected.

### 2.4. Sholl Analysis

The total dendritic branches in one GFP-positive cell in the DG were scanned by Olympus Fluoview FV1000. The branches of dendrites were analyzed by ImageJ Sholl Analysis Plugin. Five mice were selected in each group and five sections in each mouse were picked, only cells in the DG from each section were counted and analyzed.

### 2.5. BrdU Injection

Male mice were randomly divided into three groups. After MK801 administration, isoflurane treatment was applied for five consecutive days, and 5′-bromo-2′-deoxyuridine (BrdU; Sigma) in saline was administered (100 mg/kg, i.p.) twice a day (at 10:00 a.m. and 3:00 p.m.) for five days. Animals were sacrificed 24 h after the last BrdU injection.

### 2.6. Behavior Analysis

Behavioral tests: The open field test (OFT), pre-pulse inhibition (PPI) and T-maze were performed by investigators unaware of the animal genotype and grouping information. All tests were carried out during the light period, 8 to 12-week-old males were used, and mice in each group were handled for at least 15 min twice a day for 3 days prior to the behavioral tests. No body weight or motor coordination differences were found among groups during behavioral analysis.

### 2.7. Open Field Test

OFT was adopted for assessing spontaneous locomotor activity in rodent models. The mice were gently placed at the center of a square chamber (45 × 45 × 45 cm), and movements of total distance were monitored using an automated video tracking system (Supermaze 2.0, Softmaze, China) for 10 min each. The apparatus was swept with 75% alcohol to avoid the presence of olfactory cues after each trial. The total distance traveled during a session was measured.

### 2.8. Prepulse Inhibition of Acoustic Startle

PPI tests were performed in a sound-attenuated box (SR-LAB, Startle Response System, San Diego Instrument, San Diego, CA, USA). Mice were placed gently in a non-restrictive Plexiglas cylinder mounted on a plastic platform, and the motion of the mice was transduced via a piezoelectric accelerometer. Animals were allowed to habituate to the chamber prior to the test, habituating to a 70 dB background white noise for 5 min, and to 120 dB (auditory-evoked startle stimuli, 20 ms for 10 times). During the tests, mice were subjected to twelve startle trials (120 dB) and twelve prepulse/startle trials (20 ms white noise at 0, 75, 80, and 85 dB followed by 120 dB startle stimulus 20 ms at 100 ms intervals). During 100 ms of the startle stimulus onset (sampled at 1 kHz), the movement of mice was measured. The formula of PPI (%) calculation: (1 − (startle amplitude on prepulse-pulse trials/startle amplitude on pulse alone trials)) × 100%.

### 2.9. T-Maze Test

The T-maze was an enclosed apparatus with three arms, two symmetrical choice arms (30 × 7 cm), a start arm (38 × 7 cm) and flanking a central choice area (7 × 7 cm). Mice were placed gently in the start arm facing to the wall before each session. The door of the chosen one was closed for 30 s once the animal entered one of the two choice arms. The mouse was removed from the chosen arm and placed in the home cage for 1 min, then put back in the start arm to proceed the second-choice trial. The novel arm instead of the original one was the right choice in the second trial, ten sessions were applied in the second trial. Correct percentage (%) = (total entries to the novel arm in the second trial/total second trial) × 100%.

### 2.10. Immunofluorescence

Mice were anesthetized with pentobarbital sodium (50 mg/kg, i.p.) and perfused transcardially with 0.1 M PBS followed by 4% paraformaldehyde (PFA) in 0.1 M PBS. Brains were removed and fixed overnight in 4% PFA at 4 °C, then dehydrated in 30% sucrose and serially sectioned in the coronal plane at 30 µm with a cryostat (Thermo Scientific, HM550). Brain sections were blocked in blocking buffer (10% goat serum and 3% BSA in PBS with 0.3% Triton X-100 included) for 60 min at room temperature, and then incubated in primary antibodies (rabbit anti-Ki67: 1:1000, ab15580, Abcam; rat anti-BrdU 1:300, ab74545, Abcam; mouse anti-PV: 1:1000, 235, Swat; mouse anti-GFAP: 1:1000, ab7260, Abcam; rabbit anti-cFos: 1:500, ab214672, Abcam; mouse anti-NeuN: 1:1000, ab104224, Abcam) in the blocking buffer overnight at 4 °C. After washing with PBS 3 times, sections were incubated in secondary antibodies with the corresponding fluorophore-conjugated (1:500, Invitrogen) for 1 h at 37 °C. After another three washes in PBS, sections were incubated with DAPI (Beyotime Biotechnology) and mounted on glass slides with the fluorescent mounting medium. Images were taken using an Olympus Fluoview FV1000 or Zeiss LSM780 confocal microscope. The number of fluorescent cells was counted by ImageJ 1.48v (National Institutes of Health, USA).

For BrdU staining, sections were incubated with 2N HCl for 30 min at 37 °C to denature DNA, and 0.1 M borate buffer (PH8.5) for 10 min at room temperature for neutralization before incubation in blocking buffer.

### 2.11. Western Blot

Brain tissue were isolated and homogenized on ice in RIPA buffer containing 25 mM Tris-HCl (pH 7.6), 150 mM NaCl, 1% NP-40, 1% sodium deoxycholate, 0.1% SDS, 1 mM PMSF, 10 mM sodium fluoride, 2 mM sodium pyrophosphate, 1 mM sodium orthovanadate, and Roche protease phosphatase inhibitor cocktail for 30 min. Lysates were cleared with centrifugation at 13,000 rpm for 30 min at 4 °C. Protein concentrations of the supernatants were determined with a BCA protein assay kit (Beyotime Biotechnology, Shanghai, China). Protein samples were boiled in 6× SDS sample buffer for 10 min and resolved in SDS-PAGE. Proteins were transferred to the PVDF membrane (88520, Thermofisher), which were blocked in Tris buffer (pH 7.0) containing 0.1% Tween-20 and 5% BSA (blocking buffer). The membrane was incubated in primary antibodies (diluted in blocking buffer) overnight at 4 °C and washed three times before incubation with HRP-conjugated secondary antibodies (diluted in blocking buffer) for 1 h at room temperature. Protein bands were visualized with enhanced chemiluminescence substrate (RM00021, Abclonal). The band density was measured by ImageJ software, and data were analyzed by GraphPad Prism 8.0 (Graphpad Software; La Jolla, CA, USA).

Primary antibodies used were rabbit anti-GluN2A (A0924, Abclonal; 1:1000), rabbit anti-GluN2B (ab65783, Abcam; 1:1000), rabbit anti-NRG1 (A0687, Abclonal; 1:1000), rabbit anti-parrvalbumin (A2791, Abclonal; 1:1000), rabbit anti-α tubulin (AC003, Abclonal; 1:4000), and mouse anti-GAPDH (AC033, Abclonal; 1:5000).

### 2.12. Slice Electrophysiology

Mice were anesthetized with pentobarbital sodium (50 mg/kg, i.p.), and the brains were rapidly removed from the skull and sectioned in ice-cold (0–3 °C) solution containing (in mM): choline chloride 110, CaCl_2_ 0.5, KCl 2.5, MgCl_2_ 7, glucose 20, Na-ascorbate 1.3, Na-pyruvate 0.6, NaH_2_PO_4_ 1.3, NaHCO_3_ 25.0. Chilled artificial cerebrospinal fluid (ACSF) was pre-saturated with 95% O_2_ and 5% CO_2_ for 30 min. Horizontal slices (300 µm thick) were cut using a Leica VT1000S vibratome (Leica, Germany), slices were recovered at 34.5 °C for 30 min and then maintained at 25 °C in oxygenated ACSF composed of (in mM): NaCl 125, KCl 2.5, NaH_2_PO_4_ 1.3, NaHCO_3_ 25, CaCl_2_ 2, MgCl_2_ 1.3, Na-ascorbate 1.3, Na-pyruvate 0.6, glucose 10 for an additional 1 h. Subsequently, each slice was individually transferred to a submersion type recording chamber constantly super-fused with oxygenated ACSF (2 mL/min) at room temperature. Extracellular field potential recordings were obtained using microelectrodes (~1 MΩ) filled with ACSF. Field excitatory postsynaptic potentials (fEPSPs) were evoked by perforant path stimulation with a concentric bipolar tungsten electrode (World Precision Instruments). LTP was triggered with the HFS paradigm consisting of 100 pulses at 100 Hz, repeated with 20 s inter-burst interval 4 times. All data were obtained using MultiClamp 700B patch-clamp amplifiers (Molecular Devices), sampled at 5 kHz and low-pass filtered at 2 kHz using a Digidata 1550B analog-digital interface (Molecular Devices).

### 2.13. Quantification and Statistical Analysis

All statistical analyses were performed using Graphpad Prism 8.0 Software (Graphpad Software; La Jolla, CA, USA). Statistical significance was determined by Student’s *t*-Test, one-way ANOVA followed by Tukey’s post hoc test, or two-way ANOVA with Bonferroni post hoc test. Data were presented as mean ± standard error. Statistical differences were considered to be significant when *p*
*<* 0.05.

## 3. Results

### 3.1. Isoflurane Treatment Ameliorates MK801-Induced Schizophrenia-Related Behavior Phenotypes

To investigate the effect of isoflurane exposure on the MK801-induced schizophrenia model (Figure 1A), male mice were randomly selected and divided into three groups. (1) Control (Ctrl) mice subjected to saline injection. (2) MK801 mice were subjected to MK801 injection (twice a day for five days, intraperitoneally) followed by 30% oxygen treatment. (3) MK801+ISO mice were subjected to MK801 injection and isoflurane treatment (0.7% isoflurane in 30% oxygen for 1 h, once a day for 5 days) (Figure 1A). Behavioral tests were performed after five days of isoflurane exposure.

First, we evaluated hyperactivity using OFT. Five days of MK801 injection induced hyper-locomotion, and isoflurane treatment reversed the total distances traveled in the arena (Figure 1B,C). Second, the T-maze test was performed to evaluate the effect of isoflurane exposure on working memory. MK801 administration impaired working memory, which could be reversed by isoflurane treatment (Figure 1D). Third, the effect of isoflurane treatment on sensory gating was evaluated by PPI. PPI significantly increased with prepulse intensity in the control group, indicating that prepulse intensity could be distinguished by animals (Figure 1E). Moreover, the PPI attenuation induced by MK801 could also be reversed by isoflurane treatment. To exclude the toxicity of isoflurane under physiological conditions, 0.7% isoflurane in 30% oxygen was performed in wild type mice (Ctrl+ISO) for five consecutive days (Appendix A). Hence, these observations demonstrate that isoflurane treatment can rescue the MK801-induced schizophrenia-related phenotypes.

### 3.2. Isoflurane Reverses Adult Neurogenesis Deficits in the DG Induced by MK801

Ki67-positive cells were utilized as the marker of neuronal proliferation to test whether isoflurane might restore adult neurogenesis deficit caused by MK801. Administration of MK801 decreased the proliferation of neural stem cells (NSCs) in the DG; however, exposure to isoflurane reversed this effect (Figure 2B,C). Meanwhile, glial fibrillary acidic protein (GFAP), a maker of astrocytic cells was used to show that only neurons instead of glial cells were selectively generated by isoflurane treatment (Appendix A). To further elucidate the isoflurane treatment effect on the survival of NSCs, bromodeoxyuridine (BrdU) was i.p. injected for 5 days to label proliferating progenitor cells (Figure 2A). Isoflurane treatment significantly reversed BrdU+ cell numbers compared to the MK801 group (Figure 2D,E). These data suggest MK801-induced deficiencies in adult neurogenesis in the DG can be reversed by isoflurane exposure. To investigate the effect of isoflurane treatment on the neurogenesis of NSCs, we labeled the NSCs in the DG with retrovirus expressing GFP (ROV-EF1α-EGFP) by stereotaxic microinjection in the DG bilaterally (Figure 2F). After three weeks, mice were treated with MK801, followed by isoflurane exposure. MK801-induced reduction of total dendritic branches was reversed by isoflurane exposure, suggesting neurogenesis impairment could also be rescued by isoflurane (Figure 2G,H).

### 3.3. Synaptic Plasticity Deficit Induced by MK801 in the DG Can Be Reversed by Isoflurane Treatment

To measure the long-term effects of MK801 and isoflurane on glutamate transmission, we first looked at the expression of NMDA receptor subunits. In line with earlier reports, MK801 did not affect the expression of NR2A and NR2B [16,17], and neither did the subsequent isoflurane treatment, suggesting the basal glutamate transmission might not be affected (Figure 3A–C). To determine whether MK801 could impact synaptic plasticity and whether this effect could be reversed by isoflurane, as shown in Figure 3D, high-frequency stimulation (HFS) delivered to the perforant pathway (PP) to generate LTP on dentate granule cells (DGCs). Evoked field excitatory postsynaptic potentials (fEPSPs) presented a ~58% reduction in the MK801 group, consistent with previous reports [18,19], and this effect of MK801 on LTP could be partially reversed by isoflurane (Figure 3E,F), suggesting the long-term effect of MK801 on glutamate transmission could be reversed by isoflurane.

### 3.4. Isoflurane Inhalation Reverses PVIs Reduction and the Aberration of NRG1-ErbB4 Signaling in the DG Induced by MK801

The Parvalbumin (PV) positive neuron is one of the main types of GABAergic interneurons in the DG. These inhibitory neurons are known for their fast-spiking and projection to the axon initial segments or the soma of target neurons [20], and abnormalities in PVI are believed to be one of the key reason of schizophrenia symptoms [21,22]. More than 100 genes have been identified associated with schizophrenia, the majority of which are expressed in poorly functioning PV neurons and thus induce less GABA release [23]. Neuregulin 1 (NRG1) and its receptor ErbB4 are susceptibility genes of schizophrenia, and endogenous NRG1-ErbB4 signaling is critical to maintaining GABAergic activity, especially in PVIs [24,25].

Consistent with our previous observation [14], MK801 injection reduced the amount of PVIs and PV protein expression in the DG (Figure 4A,C–E), and hindered NRG1-ErbB4 signaling (Figure 4A,B,F–I). Here we found both the quantity of PVIs, and the PV expression were rescued by isoflurane exposure (Figure 4A,C–E). Similarly, the NRG1-ErbB4 signaling pathway was reversed, revealed by the expression of NRG1 and its receptor ErbB4, and the phosphorylated ErbB4 in the DG (Figure 4A,B,F–I). In summary, our data suggest that consecutive isoflurane exposure can rescue the impairment of PVI and NRG1-ErbB4 signaling, and the therapeutic effect of isoflurane in MK801-induced phenotypes might depend on the normal function of PVI in the DG.

### 3.5. The Therapeutic Effect of Isoflurane Requires PVIs in the DG

To validate whether PVI in the DG is required for the therapeutic effect of isoflurane inhalation, a Cre-dependent adeno-associated virus (AAV) expressing the proapoptotic protease caspase-3 (AAV-flex-tsCasp3-TEVp) was microinjected into the DG of PV-Cre mice to ablate the PV interneurons in the DG, and AAV-CAG-flex-eGFP was applied as control (Figure 5A). Animals were randomly divided into three groups: (1) control (Ctrl): the littermate PV-Cre mice, (2) EGFP-PV: PV-Cre mice injected with the control virus AAV-CAG-flex-eGFP, (3) Casp-PV: animals injected with AAV-flex-tsCasp3-TEVp, (4) Casp-PV-ISO. Three weeks after PV-Cre mice injected with AAV-flex-tsCasp3-TEVp, isoflurane treatment was applied for consecutive five consecutive days (Figure 5A).

Consistent with our previous observation [14], PVI ablation in the DG caused hyper-locomotion in OFT, sensory gating impairment in PPI, and working memory loss in the T-maze test (Figure 5B–E). Here we found isoflurane was unable to rescue any of these schizophrenia-related behavior phenotypes (Figure 5B–E). Meanwhile, almost no PVI in the DG was observed after AAV-flex-tsCasp3-TEVp virus injection, which cannot be rescued in Casp-PV-ISO groups (Figure 5F,G), confirming the efficacy of caspase-induced neuron ablation.

To further illustrate whether isoflurane can rescue adult neurogenesis impairment induced by PVI ablation, the number of Ki67-positive neurons and the total dendritic branches of GFP-positive neurons were observed. Consistent with our previous observation, PVI ablation in the DG induced deficits in the proliferation and maturation of newborn cells [14]. Here we found these effects could not be rescued by isoflurane (Figure 5F–I), indicating that PVI ablation imposes a restriction on adult hippocampus neurogenesis. Additionally, the number of dividing cells marked by Ki67 was also significantly reduced after PVI ablation in Casp-PV and Casp-PV-ISO groups (Figure 5F,H). These data suggest that PVIs are indispensable for reversing schizophrenia-related deficits.

Since PVI ablation induces irreversible cognitive function deficits, as shown in Figure 5, to further elucidate how PVIs mediate the therapeutic effect of isoflurane, we bilaterally injected AAV expressing enhanced green-fluorescent protein (eGFP) and tetanus toxin light chain (Tet) (AAV-DIO-EGFP-Tettox) (Figure 6A), which completely blocked evoked synaptic transmission in the DG [26]. Animals were randomly divided into three groups: (1) Control (Ctrl): the littermates PV-Cre mice, (2) PV-Tet group: PV-Cre mice injected with AAV-DIO-EGFP-Tettox, (3) PV-Tet-ISO group: PV-Cre mice injected with AAV-DIO-EGFP-Tettox and treated with isoflurane.

As expected, GABA release inhibition elicited abnormalities associated with schizophrenia, including hyper-locomotion in OFT, sensory gating deficit in PPI, and working memory impairment in the T-maze test (Figure 6B–E). Interestingly, isoflurane was able to reverse all these abnormalities (Figure 6B–E). Similarly, LTP impairment and NSCs proliferation reduction were induced by GABA release inhibition of PVIs in the DG. These effects could be reversed by isoflurane, suggesting PV GABA release might be unnecessary for the therapeutic effects of isoflurane (Figure 6F–I).

## 4. Discussion

The main results of this study are as follows. First, inhaling a low concentration of isoflurane for five days can cure the primary behavioral phenotypes of schizophrenia induced by MK801, including locomotor hyperactivity, sensory gating impairment, and working memory loss (Figure 1). Second, isoflurane treatment can restore adult neurogenesis deficits, including neurogenesis of the adult neurons and proliferation of neural stem cells in the DG (Figure 2), and the impaired long-term synaptic plasticity of the perforant path (Figure 3). Third, isoflurane therapy can reverse the downregulation of ErbB4 signaling and the decrease of PVI in the DG (Figure 4). Forth, PVIs in the DG are necessary for the isoflurane therapy; but intriguingly, the release of GABA in PVIs was not required, as demonstrated by PVI ablation and GABA release inhibition, respectively (Figure 5 and Figure 6).

Schizophrenia is characterized by positive, negative and cognitive symptoms; however, the present pharmacotherapies are ineffective for the cognitive symptoms [27]. The prefrontal cortex and the hippocampus are thought to play important roles in the cognitive symptoms, particularly in working memory [14,28]. Prior research suggests that PVI dysfunction is the key cellular pathophysiology locus in cognitive impairments of schizophrenia [29]. General perturbations of the PFC induced three symptom categories of schizophrenia, whereas only selective inactivation of PVIs caused deficits in the cognitive filed [30]. Moreover, through excitation/inhibition imbalance and deficiencies in adult neurogenesis, PVI dysfunction in the hippocampus can result in schizophrenia cognitive symptoms [31,32]. However, whether hypofunction of NMDA receptors in PVIs underlies the schizophrenia symptoms is still controversial [33,34].

In this work, schizophrenia traits were induced using the NMDA receptor antagonist, MK801. According to previous studies, MK801 has an impact on long-term synaptic plasticity [35,36]. MK801 injections at low concentration (0.033 mg/kg) for five days have been shown to enhance NR2A and NR2B expression, but injections at high concentration (1.0 mg/kg) have been shown to decrease NR2A and NR2B expression 50 min after the last dosage [37]. Given that LTP is greatly diminished seven days after the last dose of MK801 injection (0.5 mg/kg), yet NR1A and NR2B expression in the DG is unaltered (Figure 3), MK801 may suppress LTP without relying on NMDA receptors. PVIs are fast-spiking, high-metabolic-activity GABAergic neurons that are extremely vulnerable in the prefrontal cortex and hippocampus [38]. Acute MK801 administration decreases PV mRNA expression first in the DG and then in other brain regions, such as the prefrontal cortex and basolateral amygdala [39], suggesting PVIs in the DG may be more vulnerable than in other brain regions. However, acute MK801 administration promotes adult neurogenesis in the DG [40]. In contrast, repeated MK801 treatment decreases adult neurogenesis [41], which may cause the loss of PVIs in the DG [14] (Figure 5). The loss of PVI results in LTP deficit as well [42], making PVIs the primary target of therapy to reverse the behavior abnormalities induced by MK801. To investigate the initial schizophrenia-related symptoms induced by MK801, we applied MK801 i.p. injection for five consecutive days in the DG to explore how PVIs affect schizophrenia-related disorders.

Sensory gating is mediated by diverse brain regions, such as the auditory cortex, prefrontal cortex, and hippocampus [43]. The GABAergic neurons in the DG are important for sensory gating [44], but it is still unclear how these neurons influence this process. In our previous and current research, specific ablation of PVIs in the DG could impair sensory gating [14] (Figure 5), and blocking the GABA release from PVI in the DG could also decrease PPI (Figure 5 and Figure 6), indicating the significance of the GABA release from PVI in the DG in sensory gating. The primary cognitive symptom of schizophrenia is the loss of working memory, which is mainly related to the prefrontal cortex and hippocampus [45]. Working memory is significantly impacted by PVI ablation in the DG [14] (Figure 5); the same result is shown in Figure 6 after GABA release inhibition from PVIs in the DG. These findings point out the critical importance of the DG PVIs in working memory. The synchronization and spatial representations of the hippocampus depend on the NMDA receptors in the PVI [46], and PVI hypofunction in the DG may result in the imbalance between excitation and inhibition and the deficiency in adult neurogenesis [7], which in turn affects the synchrony and spatial representations of the hippocampus.

Given that isoflurane anesthesia has been linked to negative effects on cognition and adult neurogenesis [47], the concentration that we used in this study was only half of that applied during general anesthesia, allowing isoflurane-treated mice to walk around freely inside the chamber. The efficacy of isoflurane in treating schizophrenia-like phenotypes were shown by the alleviation of hyperlocomotion, PPI deficit, and spatial working memory impairment, either induced by MK801 chronic injection or the blockade of PVI-GABA release in the DG (Figure 1 and Figure 6). Likewise, the impairment LTP and adult neurogenesis induced by these two modes in the DG were reversed by isoflurane (Figure 3 and Figure 6). However, the therapeutic effect of isoflurane required the present of PVIs in the DG, as isoflurane cannot attenuate the phenotypes caused by PVI ablation in the DG (Figure 5).

Dysfunction of the prefrontal cortex is also an underlying cognitive disorder in schizophrenia, and CREB phosphorylation in the medial prefrontal cortex is increased after 20 min low-dose isoflurane treatment [48]. CREB is a critical transcription factor that modulates synaptic plasticity, and is involved in schizophrenia pathogenesis and therapy [49]. However, interestingly, low-dose and high-dose isoflurane treatments induces vasoconstriction and vasodilation, respectively, in the cerebral vessels of baboon [50] and reduced cerebral blood flow in the medial prefrontal cortex of macaque monkey [51], so whether the prefrontal cortex and the CREB signaling pathway mediate the therapeutic effect of low-dose isoflurane treatment needs further investigation. On the neural network level, a single high isoflurane exposure (1.8%) for 3 h induces prolonged enhancement (1 month) of the hippocampal-cortical connectivity, and 20 genes expression altered in the cortex are related to neuronal transmission [52], suggesting the hippocampal-cortical interaction might be involved in the isoflurane therapeutic effect. However, considering the side effects of high-dose isoflurane, further investigation is needed to illustrate the low-dose effect on hippocampal-cortical connectivity.

Similar with sevoflurane, isoflurane is also regarded as a positive-modulator of GABA_A_ receptor, increase GABA_A_ receptor current when GABA is present [53]. The GABA_A_ receptor can still tonically activate in the absence of GABA [54], while, the tonic inhibition mediated by extra-synaptic GABA_A_ receptor is the main target for anesthesia [55]. Once PVI-GABA release is blocked by tetanus toxin in the DG (Figure 6), isoflurane might compensate for the inhibition effect on the glutamatergic neurons through tonic activation of GABA_A_ receptor. In contrast to tetanus toxin, the expression of caspase-3 eliminates PVIs in the DG, while the schizophrenia-like phenotypes cannot be attenuated by isoflurane (Figure 5), suggesting that in the absence of GABAergic transmission, PVIs presentation in the DG is required for isoflurane therapeutic effects. NRG1-ErbB4 signaling is required to maintain the normal function of PVIs [25,56,57]. ErbB4 is highly expressed in PVI, while isoflurane treatment rescued the reduction of NRG1-ErbB4 expression induced by MK801 injection (Figure 4), which may contribute to the effects of isoflurane working on glutamatergic neurons through NRG1-ErbB4 downstream pathways or the retrograde pathways. It has been reported that NRG1 retrograde signaling affects glutamatergic neurons through its intracellular segment binding with LIMK1 [58]. Therefore, further investigation is needed to explore the function of the survival of PVIs in isoflurane therapy.

## 5. Conclusions

Our results show that low concentrations of isoflurane exposure can reduce MK801-induced locomotor hyperactivity, improving PPI deficits and working memory loss, as well as adult neurogenesis and synaptic plasticity impairments in the DG We showed that isoflurane alleviates schizophrenia-related disorders through the presence of PVIs in the DG, providing a new strategy to treat schizophrenia.

## Figures and Tables

**Figure 1 biomedicines-10-02759-f001:**
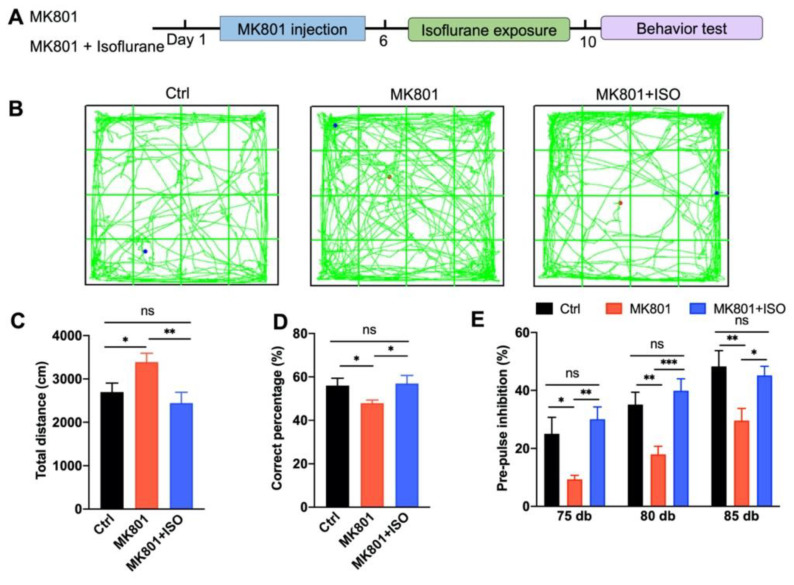
MK801-induced schizophrenia-related behavioral phenotypes are reversed by isoflurane. (**A**) Schematic diagram of the experimental design. After 5 days of MK801 injection and 5 days of isoflurane treatment, mice were subjected to behavioral tests. (**B**) Representative traces of mice in OFT. (**C**) Isoflurane treatment reversed the hyper-locomotion phenotype induced by MK801, revealed by OFT (*n* = 10, 11, and 10 mice in Ctrl, MK801, and MK801+ISO groups, respectively. One-way ANOVA, *F*_(2, 28)_ = 5.12, *p* = 0.01; post hoc test: Ctrl vs. MK801, *p* = 0.02; MK801 vs. MK801+ISO, *p* = 0.007; Ctrl vs. MK801+ISO, *p* = 0.46). (**D**) The working memory deficit induced by MK801 was attenuated by isoflurane exposure (*n* = 9, 11, and 10 mice in Ctrl, MK801, and MK801+ISO groups, respectively. One-way ANOVA, *F*_(2, 27)_ = 2.726, *p* = 0.01; post hoc test: Ctrl vs. MK801, *p* = 0.04; MK801 vs. MK801+ISO, *p* = 0.03; Ctrl vs. MK801+ISO, *p* = 0.84). (**E**) Isoflurane attenuated the pre-pulse inhibition deficit induced by MK801 (*n* = 10, 14, and 12 mice in Ctrl, MK801, and MK801+ISO groups, respectively. Two-way ANOVA, *F*_(2, 98)_ = 18.11, *p* < 0.0001; post hoc test: Ctrl vs. MK801, *p* = 0.004; MK801 vs. MK801+ISO, *p* = 0.01; Ctrl vs. MK801+ISO, *p* = 0.72). Data are represented as mean ± SEM, * *p* < 0.05, ** *p* < 0.01, *** *p* < 0.001, ns, not significant.

**Figure 2 biomedicines-10-02759-f002:**
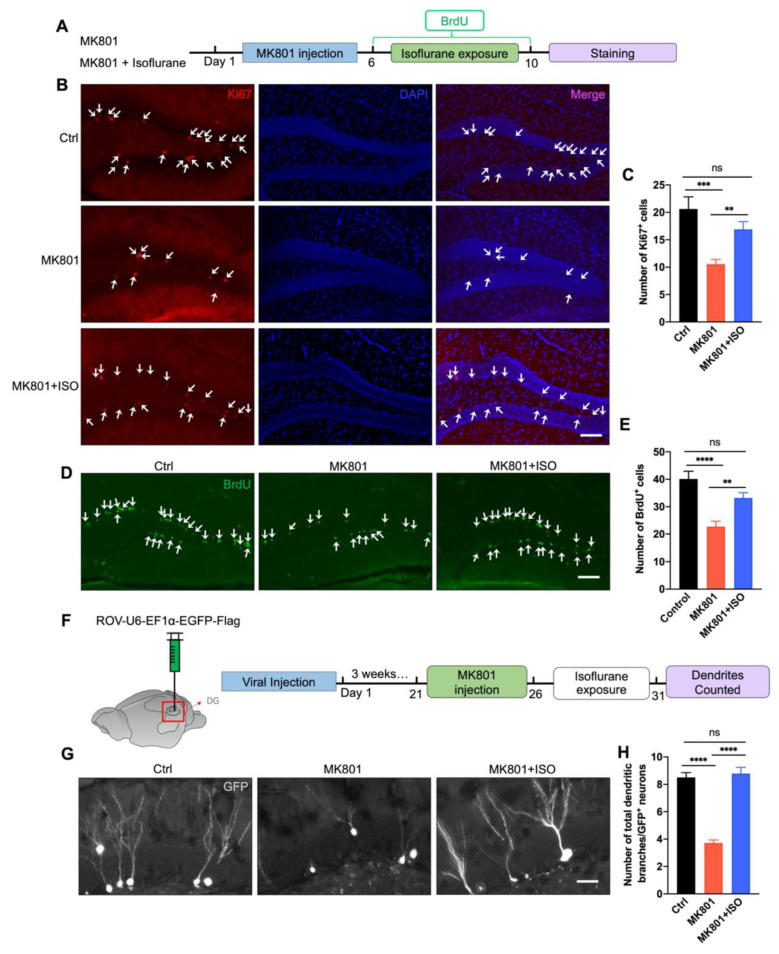
Isoflurane treatment reverses NSCs adult neurogenesis deficit in the DG induced by MK801. (**A**) Schematic of BrdU injection. BrdU was injected twice a day during 5 days of isoflurane exposure. (**B**) Representative photomicrographs showing Ki67-positive cells in the DG, indicated by white arrows. Scale bar, 100 μm. (**C**) Isoflurane inhalation reversed the reduction of Ki67-positive cells in the DG induced by MK801 (five mice in each group and five sections in each animal were picked and counted. One-way ANOVA, *F*_(2, 29)_ = 9.306, *p* = 0.0008; post hoc test: Ctrl vs. MK801, *p* = 0.0005; MK801 vs. MK801+ISO, *p* = 0.009; Ctrl vs. MK801+ISO, *p* = 0.08). (**D**) Representative photomicrographs showing BrdU-positive cells in the DG, indicated by white arrows. Scale bar, 100 μm. (**E**) Isoflurane inhalation reversed the reduction of BrdU+ cells in the DG induced by MK801 (five mice in each group and five sections in each animal were picked and counted. One-way ANOVA, *F*_(2, 28)_ = 15.33, *p* < 0.0001; post hoc test: Ctrl vs. MK801, *p* < 0.0001; MK801 vs. MK801+ISO, *p* = 0.0012; Ctrl vs. MK801+ISO, *p* = 0.051). (**F**) Schematic of retrovirus injection. Retrovirus was micro-injected into the DG 21 days before MK801 injection. (**G**) Representative images of the dendrites of newborn dentate granular cells. Scale bar, 10 μm. (**H**) The reduction of dendritic branches of newborn dentate granular cells reversed by isoflurane treatment (five mice in each group and five sections in animal were picked, and three cells of each section were analyzed and counted. One-way ANOVA, *F*_(2, 39)_ = 62.68, *p* < 0.0001; post hoc test: Ctrl vs. MK801, *p* < 0.0001; MK801 vs. MK801+ISO, *p* < 0.0001; Ctrl vs. MK801+ISO, *p* = 0.62). Data are represented as mean ± SEM, ** *p* < 0.01, *** *p* < 0.001, **** *p* < 0.0001, ns, not significant.

**Figure 3 biomedicines-10-02759-f003:**
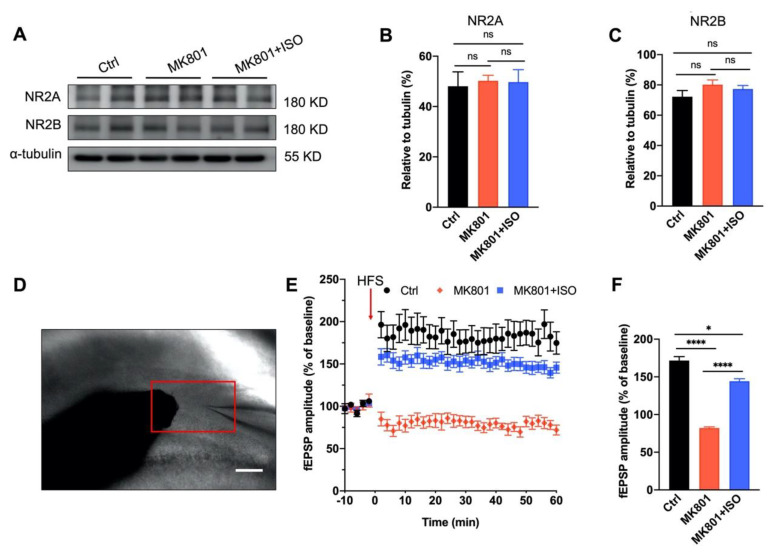
MK801-induced synaptic plasticity deficit in the DG can be reversed by isoflurane treatment. (**A**–**C**) Representative Western blotting images of proteins (**A**) and densitometric quantification (**B**,**C**) for NR2A and NR2B in the DG. α-tubulin was used as the loading control (*n* = 3 mice per group. One-way ANOVA, *F*_(2, 9)_ = 9.306, *p* = 0.062; post hoc test: Ctrl vs. MK801, *p* = 0.73; MK801 vs. MK801+ISO, *p* = 0.92; Ctrl vs. MK801+ISO, *p* = 0.83 shown on B; One-way ANOVA, *F*_(2, 9)_ = 1.588, *p* = 0.256; post hoc test: Ctrl vs. MK801, *p* = 0.16; MK801 vs. MK801+ISO, *p* = 0.48; Ctrl vs. MK801+ISO, *p* = 0.31 shown on (**C**)). (**D**) Electrode configuration for the LTP recording in the DG. The red rectangle indicates the stimulating electrode (left) and recording electrode (right). Scale bar, 100 μm. (**E**) LTP of local field potential in the DG after HFS induction at the perforant path. LTP was enhanced in the DG by isoflurane treatment after MK801 injection (*n* = 5 mice per group). (**F**) Quantitative analysis of data in (**E**). (Two-way ANOVA, *F*_(2, 1260)_ = 713.5, *p* < 0.0001; post hoc test: Ctrl vs. MK801, *p* < 0.0001; MK801 vs. MK801+ISO, *p* < 0.0001; Ctrl vs. MK801+ISO, *p* < 0.05). Data are represented as mean ± SEM, * *p* < 0.05, **** *p* < 0.0001, ns, not significant.

**Figure 4 biomedicines-10-02759-f004:**
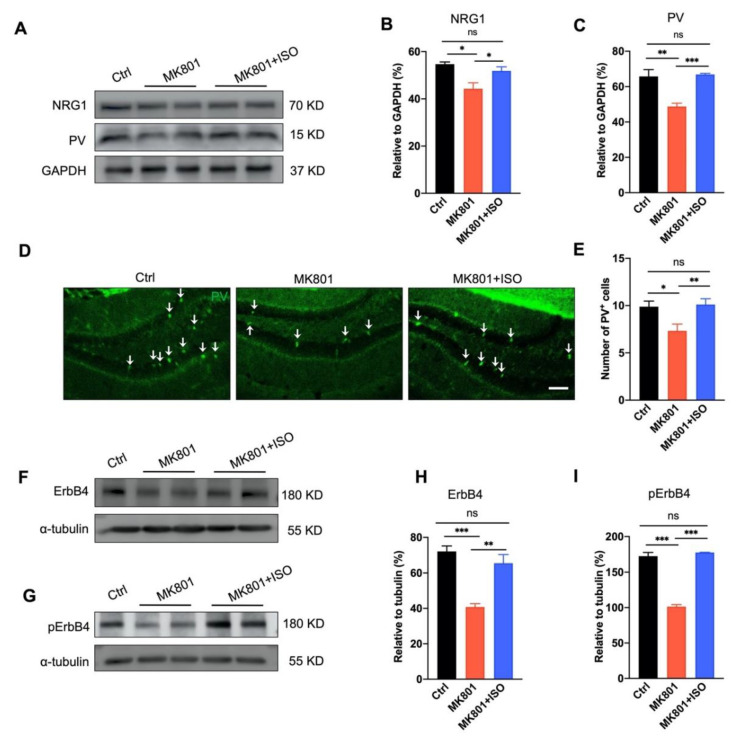
Isoflurane reverses MK801’s effects on PVIs reduction and NRG1-ErbB4 signaling down-regulation in the DG. (**A**) Representative Western blotting image of NRG1 and PV in the DG. GAPDH served as loading control. (**B**,**C**) Quantification of NRG1 and PV expression in A. Isoflurane reversed MK801 induced NRG1 and PV reduction in the DG (*n* = 3 mice per group. one-way ANOVA, *F*_(2, 9)_ = 8.508, *p* = 0.0084; post hoc test: Ctrl vs. MK801, *p* = 0.0082; MK801 vs. MK801+ISO, *p* = 0.048; Ctrl vs. MK801+ISO, *p* = 0.19 shown on B; one-way ANOVA, *F*_(2, 9)_ = 16.71, *p* = 0.0009; post hoc test: Ctrl vs. MK801, *p* = 0.0072; MK801 vs. MK801+ISO, *p* < 0.0001; Ctrl vs. MK801+ISO, *p* = 0.78 shown on (**C**)). (**D**) Representative photomicrographs showing PV-positive cells in the DG. Scale bar, 100 μm. (**E**) Quantification of PV neuron number in (**D**). Isoflurane reversed MK801 induced the reduction of PV-positive neuron number in the DG (five mice in each group and five sections in each animal were picked, and three cells of each section were analyzed and counted. One-way ANOVA, *F*_(2, 24)_ = 5.863, *p* = 0.0084; post hoc test: Ctrl vs. MK801, *p* = 0.013; MK801 vs. MK801+ISO, *p* = 0.009; Ctrl vs. MK801+ISO, *p* = 0.796). (**F**,**G**) Representative Western blotting images of ErbB4 and phosphorylated-ErbB4 in the DG. α-tubulin served as loading control (*n* = 3 mice per group). (**H**,**I**) Quantification data of ErbB4 and phosphorylated-ErbB4 expression in the DG. Isoflurane reversed MK801 induced the reduction of ErbB4 and phosphorylated-ErbB4 in the DG. One-way ANOVA, *F*_(2, 9)_ = 22.48, *p* = 0.0003; post hoc test: Ctrl vs. MK801, *p* = 0.0001; MK801 vs. MK801+ISO, *p* = 0.001; Ctrl vs. MK801+ISO, *p* = 0.298 shown on (**H**); One-way ANOVA, *F*_(2, 9)_ = 8152.9, *p* < 0.0001; post hoc test: Ctrl vs. MK801, *p* < 0.0001; MK801 vs. MK801+ISO, *p* < 0.0001; Ctrl vs. MK801+ISO, *p* = 0.3566 shown on (**I**). Data are represented as mean ± SEM, two-tailed Student’s *t*-test, * *p* < 0.05, ** *p* < 0.01, *** *p* < 0.001, ns, not significant.

**Figure 5 biomedicines-10-02759-f005:**
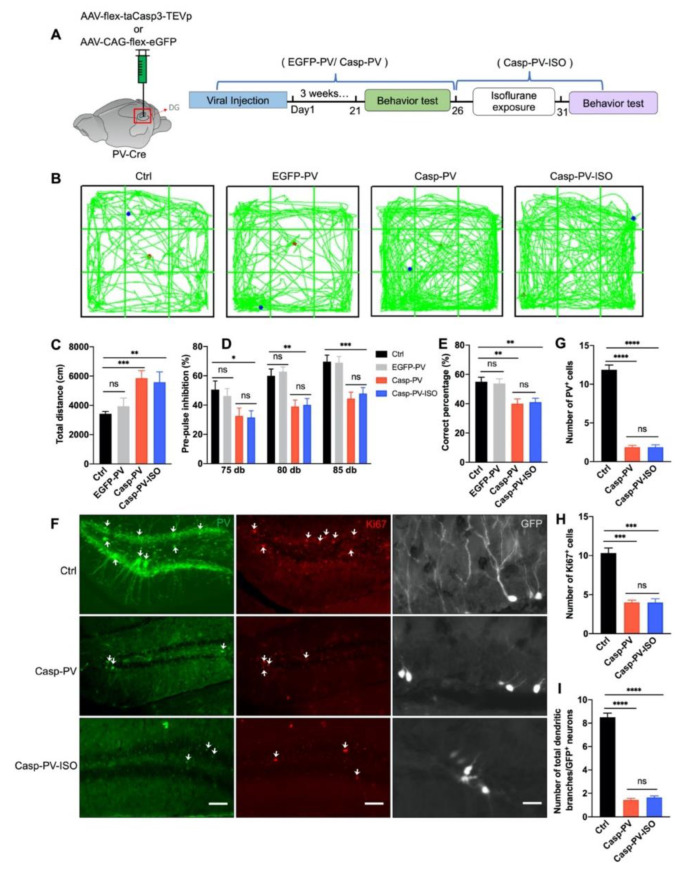
Isoflurane cannot improve schizophrenia-like phenotypes and neurogenesis induced by PV neurons ablation in the DG. (**A**) PV-cre mice were stereotaxically injected with AAV-flex-tsCasp3-TEVp (Casp-PV) to ablate PV neurons in the DG, AAV-CAG-flex-eGFP injected mice (EGFP-PV) was used as control. Schematic experimental design is shown in the right panel. (**B**) Representative traces of mice in the OFT. (**C**) Isoflurane treatment could not reverse the hyper-locomotion phenotype after PV ablation, as revealed by OFT (*n* = 10, 10, 9 and 10 mice in Ctrl, EGFP-PV, Casp-PV, and Casp-PV-ISO groups, respectively. One-way ANOVA, *F*_(3, 25)_ = 5.737, *p* = 0.004; post hoc test: Ctrl vs. EGFP-PV, *p* = 0.37; Ctrl vs. Casp-PV, *p* = 0.0003; Ctrl vs. Casp-PV-ISO, *p* = 0.007; Casp-PV vs. Casp-PV-ISO, *p* = 0.75). (**D**) Isoflurane could not rescue the pre-pulse inhibition deficit induced by PV neurons ablation (*n* = 10, 10, 9 and 10 mice in Ctrl, EGFP-PV, Casp-PV, and Casp-PV-ISO groups, respectively. Two-way ANOVA, *F*_(2, 105)_ = 14.69, *p* < 0.0001; post hoc test: Ctrl vs. EGFP-PV, *p* = 0.98; Ctrl vs. Casp-PV, *p* = 0.02; Ctrl vs. Casp-PV-ISO, *p* = 0.003; Casp-PV vs. Casp-PV-ISO, *p* = 0.8). (**E**) The working memory deficit could not be alleviated by isoflurane once PV ablation (*n* = 9, 9 and 10 mice in Ctrl, EGFP-PV, Casp-PV, and Casp-PV-ISO groups, respectively. One-way ANOVA, *F*_(3, 31)_ = 6.915, *p* = 0.0011; post hoc test: Ctrl vs. EGFP-PV, *p* = 0.78; Ctrl vs. Casp-PV, *p* = 0.004; Ctrl vs. Casp-PV-ISO, *p* = 0.0034; Casp-PV vs. Casp-PV-ISO, *p* = 0.79). (**F**) Representative photomicrographs showing PV-positive (left panel; scale bar, 100 μm), Ki67-positive neurons (middle panel; scale bar, 100 μm) and dendritic branches of NSCs (right panel; scale bar, 10 μm) after PV neuron ablation. (**G**) Quantification of PV-positive neuron number in (**F**). Isoflurane could not rescue the reduction of PV-positive neurons in the DG (five mice in each group, and five sections were picked in each mouse. One-way ANOVA, *F*_(2, 21)_ = 5.737, *p* < 0.0001; post hoc test: Ctrl vs. Casp-PV, *p* < 0.0001; Ctrl vs. Casp-ISO, *p* < 0.0001; Casp-PV vs. Casp-PV-ISO, *p* > 0.99). (**H**) Quantification of Ki67-positive neuron number in (**F**). Isoflurane could not rescue the reduction of Ki67-positive cells in the DG (five mice in each group, and five sections were picked in each mouse. One-way ANOVA, *F*_(2, 24)_ = 5.55, *p* < 0.0001; post hoc test: Ctrl vs. Casp-PV, *p* < 0.0001; Ctrl vs. Casp-PV-ISO, *p* < 0.0001; Casp-PV vs. Casp-PV-ISO, *p* > 0.99). (**I**) The reduction of dendritic branches of newborn dentate granular cells could not be reversed by isoflurane treatment (five mice in each group, and three sections were picked and counted in each mouse. One-way ANOVA, *F*_(2, 39)_ = 293.1, *p* < 0.0001; post hoc test: Ctrl vs. Casp-PV, *p* < 0.0001; Ctrl vs. Casp-PV-ISO, *p* < 0.0001; Casp-PV vs. Casp-PV-ISO, *p* = 0.27. Data are represented as mean ± SEM, * *p* < 0.05, ** *p* < 0.01, *** *p* < 0.001, **** *p* < 0.0001, ns, not significant).

**Figure 6 biomedicines-10-02759-f006:**
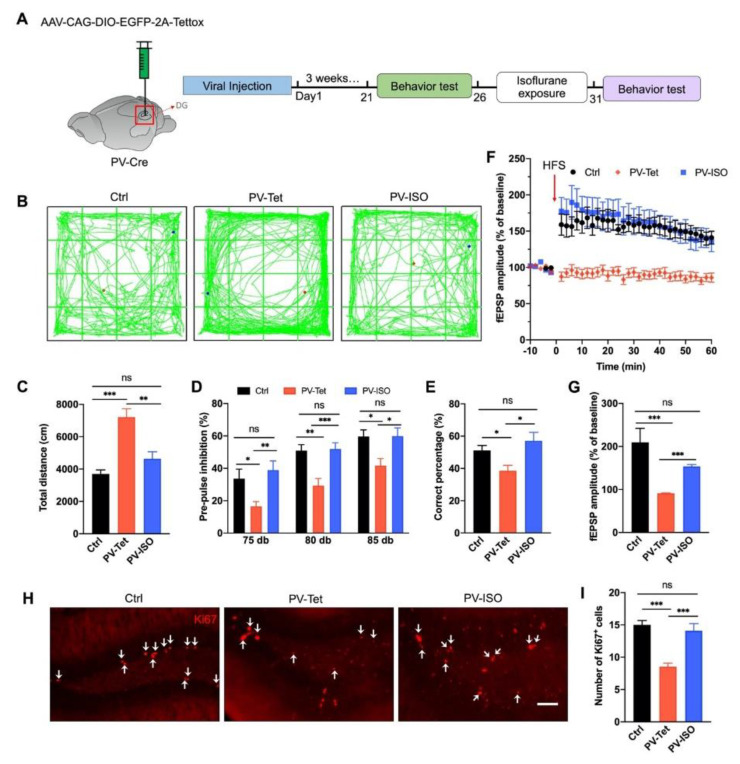
Isoflurane improves schizophrenia-related phenotypes induced by PV neuron’s GABA release inhibition in the DG. (**A**) AAV-CAG-DIO-EGFP-2A-Tettox was injected into the DG of PV-Cre mice to inhibit GABA release. (**B**) Representative traces of mice in the OFT. (**C**) Isoflurane treatment reversed the hyper-locomotion phenotype induced by PV neuron’s GABA release inhibition, revealed by OFT (*n* = 8, 10, and 8 mice in control, PV-Tet, and PV-ISO groups, respectively. One-way ANOVA, *F*_(2, 23)_ = 18.24, *p* < 0.0001; post hoc test: Ctrl vs. PV-Tet, *p* < 0.0001; PV-Tet vs. PV-Tet, *p* = 0.002; Ctrl vs. PV-ISO, *p* = 0.079). (**D**) Isoflurane attenuated the pre-pulse inhibition deficit induced by PV positive neuron’s GABA release inhibition (*n* = 9, 10, and 10 mice in Ctrl, PV-Tet, and PV-ISO group, respectively). Two-way ANOVA, *F*_(2, 72)_ = 2.873, *p* < 0.0001, post hoc test: Ctrl vs. PV-Tet, *p* = 0.01; PV-Tet vs. PV-ISO, *p* = 0.008; Ctrl vs. PV-ISO, *p* = 0.49). (**E**) The working memory deficit induced by PV neuron’s GABA release inhibition could be attenuated by isoflurane exposure (*n* = 9, 10, and 10 mice in Ctrl, PV-Tet, and PV-ISO group, respectively. One-way ANOVA, *F*_(2, 19)_ = 5.683, *p* = 0.01; post hoc test: Ctrl vs. PV-Tet, *p* = 0.01; PV-Tet vs. PV-ISO, *p* = 0.01; Ctrl vs. PV-ISO, *p* = 0.32). (**F**) Isoflurane reversed HFS induced LTP deficit induced by PV neuron’s GABA release inhibition (*n* = 5 mice per group). (**G**) Quantitative analysis of data in (**F**). (Two-way ANOVA, *F*_(2, 1155)_ = 713.5, *p* = 0.0002; post hoc test: Ctrl vs. PV-Tet, *p* = 0.0007; PV-Tet vs. PV-ISO, *p* < 0.0001; Ctrl vs. PV-ISO, *p* = 0.11). (**H**) Representative photomicrographs showing Ki67-positive cells in the DG. White arrowheads indicate target cells. Scale bar, 100 μm. (**I**) Quantification data of (**H**). Five mice in each group, and five sections were picked and counted in each mouse. One-way ANOVA, *F*_(2, 24)_ = 23.59, *p* < 0.0001; post hoc test: Ctrl vs. PV-Tet, *p* < 0.0001; PV-Tet vs. PV-ISO, *p* < 0.0001; Ctrl vs. PV-ISO, *p* = 0.5034. Data are represented as mean ± SEM. * *p* < 0.05, ** *p* < 0.01, *** *p* < 0.001, ns, not significant.

## Data Availability

Not applicable.

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
