# Peer review of "Isoflurane Rescue Schizophrenia-Related Deficits through Parvalbumin-Positive Neurons in the Dentate Gyrus"

_biomedicines, 2022, doi:10.3390/biomedicines10112759_

Round 1
Reviewer 1 Report
The present manuscript was well organized and interest to readers.
Some minor concerns should be improved and responded to reviewer's comments.
In Materials and Methods, "Mice (6-week-old)" in 2.2. and "Adult mice" in 2.3. were used, and "Only 8 to 12-week-old males were used" in 2.6. Behavioral Analysis. How old were adult mice in 2.3.? Moreover, if 6-week-old mice were subjected to the experimental design and drug administration described in 2.2., did the mice reach 8-week-old?
In Results 3.1., why did not the authors study Control+ISO group?
In Figure 2, were number of Ki67-positive cells (C) and number of BrdU-positive cells (E) counted in 1 section or image area?
In 3.3., the authors wrote "to measure the long-term effects" in text. What did "long-term effects" mean? Was treatment period 5 days long-term?
In Figures 3 and 4, the irrelevant bands in western blot images would be better to delete.
In 3.6., was it confirmed that GABA release was completely blocked in PV-Tet-ISO group similar to PV-Tet groups?
In Discussion, page 16, line 18, correct "Figure 5" to "Figure 6"; in the sentence "PVI-GABA release is blocked by tetanus toxin in the DG".
In References, [49] and [52] should be corrected and completed.
Reviewer 2 Report
The manuscript entitled „Isoflurane rescues schizophrenia-related deficits through parvalbumin –positive neurons in the dentate gyrus” by Peng et al. deals with an important issue of possible treatments of schizophrenia and neurobiological basis of the treatment. In the manuscript authors showed that low doses of isoflurane can alleviate schizophrenia symptoms and that parvalbumin interneurons in the dentate gyrus are crucial for this phenomenon. The authors presented nice data with coherent story. However, I feel that some additions could be done to improve the manuscript and the results.
1. Although authors put schemes of experimental setting in the images I feel that it would be useful to put similar description with all experiments described in the material and methods section.
2. When explaining BrdU experiments a more detailed description of groups used would be useful.
3. In results section I believe that an addition of one more group (control + isoflurane) would be useful to better asses’ value of isoflurane. For example is hyper-locomotion reduction exclusive to MK801 or something similar can be observed in control, etc.
4. Reduction of dendritic branching is not necessary a problem of NSCs maturation. There are also other reasons why this could happen. This statement in results need more evidence or rephrasing.
5. Ki67 is general marker of mitosis. Isoflurane could increase glial proliferation. An assessment of number of neurons and glial cell in DG of MK801 after isoflurane should be done to show that neurons were selectively generated.
6. I think that mechanisms in paragraph 3.5. and 3.6. should be presented together in order to get a more detailed and clear picture of what PVI are doing and what is the role of GABAergic transmission in these processes?
7. In discussion part I believe that authors should discuss the impact of hippocampus on schizophrenia vs. prefrontal cortex and possible influence of isoflurane of prefrontal cortex (important for humans).
Round 2
Reviewer 2 Report
In current form manuscript is acceptable.